Unbiased machine learning-assisted approach for conditional discretization of human performances

Banditwattanawong Thepparit 1
Masdisornchote Masawee 2 masawee.ma@spu.ac.th
1 Department of Computer Science, Faculty of Science, Kasetsart University , Krung Thep Maha Nakhon , Thailand
2 School of Information Technology, Sripatum University , Krung Thep Maha Nakhon , Thailand
Cirillo Stefano
Electronic publication date: 2025 Apr 21
Publication date: 2025
Volume: 11
Electronic Location ID: e2804
Received 2024 Aug 23; Accepted 2025 Mar 13
Copyright: © 2025 Banditwattanawong and Masdisornchote
Copyright year: 2025
Copyright holder: Banditwattanawong and Masdisornchote
License: This is an open access article distributed under the terms of the Creative Commons Attribution License, which permits unrestricted use, distribution, reproduction and adaptation in any medium and for any purpose provided that it is properly attributed. For attribution, the original author(s), title, publication source (PeerJ Computer Science) and either DOI or URL of the article must be cited.
License URL: https://creativecommons.org/licenses/by/4.0/

Keywords: Data analysis, Human performance, Conditional performance discretization, Norm referenced evaluation, Unbiasedness, Multi-modal technique, Clustering technique, Unsupervised learning, Heuristic method

Funding: Department of Computer Science and International SciKU Branding (ISB), Faculty of Science, Kasetsart University, Thailand This work was financially supported by Department of Computer Science and International SciKU Branding (ISB), Faculty of Science, Kasetsart University, Thailand. There was no additional external funding received for this study. The funders had no role in study design, data collection and analysis, decision to publish, or preparation of the manuscript.

==============================
Performance discretization maps numerical performance values to ordinal categories or performance ranking labels. Norm-referenced performance discretization is extensively applied in human performance evaluation such as grading academic achievements and determining salary increases for employees. These tasks stipulate a common condition that certain performance ranking labels might have no associated performance values and are referred to as conditional discretization. Currently, the only statistical method available for norm-referenced performance discretization is Z score, which merely addresses partial conditions. To achieve a fully conditionally norm-referenced performance discretization, this article proposes four novel approaches enlisting a multi-modal technique that incorporates unsupervised machine-learning algorithms and a heuristic method as well as a novel decision function ensuring conditional unbiasedness. The machine-learning-based methods demonstrate superiority over the heuristic one across most testing data sets, achieving a conditional unbiasedness degree ranging from 0.11 to 0.82. On the other hand, the heuristic method notably outperforms for a specific data set, exhibiting a conditional unbiasedness degree up to 0.76. Leveraging the strengths of these constituent methods enable the effectiveness of the proposed multi-modal approach for conditionally norm-referenced performance discretization.

Introduction

In pursuit of performance improvement objectives, the assessment of current performance is an indispensable practice across a human resource domain, including employee performance evaluation, talent competition, standardized tests, and educational grading. Ensuring the interpretability of performance evaluations for stakeholder understanding is paramount. Two principle schemes for human-interpretable performance evaluation are criterion-referenced and norm-referenced schemes (Wadhwa, 2008). The criterion-referenced scheme translates measured performance into absolute rating labels according to a predetermined rubric. For instance, any performance score falling between 90% and 100% is labelled as excellent, A, etc. On the other hand, the norm-referenced scheme converts performance values into relative ranking labels, depicting performance in comparison to peers rather than comparison against predefined standards. For instance, a score 85 out of 100 is assigned labels 90 percentile, high, AA, etc. Unlike the former scheme, the performance ranking labels derived via the latter scheme inherently reflects the relative quality of individual performances within a group and serves as the primary focus of this article.

One fundamental characteristic of norm-referenced scheme (also shared by criterion-referenced one) is that some performance ranking labels are not assigned in the absence of corresponding performance values. This characteristic is termed as conditional performance discretization (CPD) as opposite to unconditional performance discretization (UPD). For CPD example, in the grading of test takers’ performances using A, B, and C ranking labels, grade A might not be assigned if no test taker achieves a relatively high level of qualitative performance. A conventional method that supports norm-referenced CPD is Z score (aka normalized T score) (Wadhwa, 2008). This method statistically quantifies how far a particular performance value is from the mean of a group of performance values. It is expressed in terms of standard deviations from the mean, for instance, performance values within one standard-deviation range of the mean may be considered average, while performances above or below that range may be regarded as above or below average. Z-score method does not assign a ranking label for a specific range of standard deviations that lacks a corresponding performance value. However, Z-score method always assigns the first and the last ranking labels since the performance value intervals using Z-score method are derived from the maximum and minimum t scores rather than the upper and lower bounds of the t scores. Thus, Z-score method only partially accommodates CPD.

Importantly because norm-referenced scheme has no predefined criteria agreed upon by all stakeholders, using it for CPD is susceptible to bias. To address this ethical problem, this study firstly formalizes CPD and subsequently proposes the set of requirements to attain unbiasedness in norm-referenced CPD. The requirements are consequently utilized to derive a novelly conditional unbiasedness metric for measuring the effectiveness of norm-referenced CPD methods. Furthermore, this article proposes CPD methods leveraging heuristic, machine learning, and multi-modal approaches. By employing the conditional unbiasedness metric, the proposed approaches are compared across various real-world data sets acquired from target domains. In summary, the contributions of this article are eight folds: We propose the formal definition of CPD, as presented in Eq. (1), to establish a clear and systematic basis for introducing novel algorithms and performance metrics under a specific condition where a set of performance-ranking labels assigned is the proper subset of a set of assignable performance-ranking labels.

We introduce original unbiasedness requirements for norm-referenced CPD, establishing the foundation for novel performance metrics and the development of new norm-referenced CPD methods.

We propose four distinct methods for norm-referenced CPD that are exclusively based on heuristic, machine learning, and multi-modal approaches.

We propose a performance metric for quantifying the degrees of conditional unbiasedness that are attained by norm-referenced CPD approaches.

We originally report and discuss the findings of unbiasedness comparison among the proposed methods and a conventional one.

While this article focuses on norm-referenced CPD in the context of human performance evaluation (e.g., employee and student assessments), the proposed methods rely on no domain-specific assumption making it adaptable to a wide range of applications in other domains including healthcare (e.g., discretizing patient health metrics like body mass index (BMI) or blood pressure into rank-based categories for population health studies or risk stratification), finance (e.g., ranking financial portfolios or credit scores into percentile-based tiers for benchmarking and risk assessment), sports analytics (e.g., classifying athlete performance metrics like speed or accuracy into skill levels for talent scouting or training optimization), and environmental science (e.g., categorizing pollution levels or climate data into severity bands for comparative analysis across regions).

The rest of this article is organized as follows: Related Work explores related work. Definition of Conditional Performance Discretization section formalizes the definition of CPD. Section Conditional Unbiasedness Requirements and Metric for Norm-Referenced CPD provides conditional unbiasedness requirements and a metric for CPD. Norm-Referenced CPD Methodology section describes four novel norm-referenced CPD methods. Section Evaluation evaluates the proposed methods against various data sets and a conventional method. Section Finding and Discussion discusses findings. Last but not least, section Conclusion concludes important findings and states future research.

Related work

In the broader context of various domains, existing discretization methods (Baron & Stańczyk, 2020; García et al., 2013; Dash, Paramguru & Dash, 2011) were designed to transform continuous data into nominal data either during the preprocessing or postprocessing stages of data mining. These methods are categorized as supervised or unsupervised schemes. Supervised algorithms incorporate class information, often in the form of entropy, to determine cut-points to partition data space. In contrast, unsupervised algorithms do not consider class information. Instead, they may evenly divide the data range into a specified number of bins (namely equal-width method) or allocate an equal number of continuous values into each bin (namely equal-frequency-binning method). Yang & Webb (2008) proposed unsupervised discretization methods that aimed to reduce classification bias or error. However, all of existing discretization methods predominantly prioritized the performance enhancement of general data classification rather than ensuring unbiasedness in the context of human performance.

Extensive research in human resource management evaluated personnel’s overall performance by classifying various performance factors into a set of predefined performance ranking labels (e.g., excellent, good, regular, and insufficient). The classification tasks leveraged machine-learning classifiers (e.g., decision tree, k-nearest neighbors, support vector machine, naive bayes, and artificial neural network) and/or fuzzy logic using class labels created via either norm- or criterion-referenced methods. However, these efforts primarily focused on the discretization of multiple raw performance values rather than single inclusive performance values. Furthermore, some of these studies were conducted without addressing bias issues such as Sharma & Goyal (2015), Pan (2021), and Ahned & Sultana (2013) whereas others were aware of biasedness at the level of either human performance indicators or algorithmic operations such as Nayem & Uddin (2024) and De Oliveira Góes & De Oliveira (2020). Apart from the classification-based discretization, there is limited research (Dang, Truong & Huynh, 2021; Huang & Wang, 2022; Prasad, Choudhary & Ankayarkanni, 2020) employing clustering techniques to identify overall similarities among multiple raw performance factors and align them with performance ranking labels to gain insights into personnel performances and enhance human resource management. Unfortunately, these clustering-based discretization approaches overlooked the aspect of unbiasedness. Furthermore, these research endeavors both classification-based and clustering-based entirely engaged UPD instead of CPD.

Prior norm-referenced performance discretization methods are primarily utilized in the field of education, particularly in scenarios where exams cannot comprehensively assess all learning topics due to time constraints and limited grading resources. In such cases, norm-referenced approaches are preferred over criterion-referenced schemes (Banditwattanawong & Masdisornchote, 2021). These methods include Arora & Badal (2013), Borgavakar & Shrivastava (2017), Parveen, Alphones & Naz (2017), Shankar et al. (2016), Xi (2015), Iqbal et al. (2019), Ramen & Joachims (2014), and Bai & Chen (2006). They mostly utilized K-means clustering as reviewed in previous work (Banditwattanawong, Jankasem & Masdisornchote, 2023). Recently, Omar, Alzahrani & Zohdy (2020) leveraged K-means and the elbow to partition students’ performance data, consisting of course name, course grade, and cumulative GPA, into groups for developing improvement plans. Pamungkas, Dewi & Putri (2024) also utilized K-means and the elbow method to cluster students’ GPAs, credits taken, and the number of poor grade symbols into three groups: high achievers, average performance, and needs improvement, for developing challenging or remedial programs. In previous studies (Banditwattanawong & Masdisornchote, 2020a, 2020b, 2021), we proposed a heuristic approach for norm-referenced grading and were the first to demonstrate the application of K-means and partitioning around medoids (PAM) for this purpose. We evaluated the grading quality of the heuristic, Z-score, K-means, and PAM based on Davies-Bouldin index without the notion of fairness. Our seminal article (Banditwattanawong, Jankasem & Masdisornchote, 2023) proposed a norm-referenced achievement grading method that focused on grading fairness. This approach is, however, limited to UPD. In summary, none of these norm-referenced educational assessments aimed for CPD unbiasedness, and major changes must be applied to these studies to achieve CPD.

Table 1 summarizes these related works and their shortage in addressing our research problem of unbiasedness-centric CPD.

Table 1 Summary of related work.

Related work	Key contribution	Aware of fairness	Support for norm-referenced CPD	
Baron & Stańczyk (2020)	Comparative evaluation of ranking-based discretization methods	No	No	
García et al. (2013)	Comparative evaluation of supervised and unsupervised discretization methods	No	No	
Dash, Paramguru & Dash (2011)	Comparative analysis of discretization methods	No	No	
Yang & Webb (2008)	Unsupervised machine-learning discretization methods	No	No	
Sharma & Goyal (2015)	Machine-learning models for predicting employee performance	No	No	
Pan (2021)	Classification models for human performance based on a criterion-referenced scheme	No	No	
Ahned & Sultana (2013)	A fuzzy logic approach for employee performance evaluation	No	No	
Nayem & Uddin (2024)	Machine-learning models for predicting employee performance	Yes	No	
De Oliveira Góes & De Oliveira (2020)	A fuzzy-logic and machine-learning based approach for human resource performance evaluation	Yes	No	
Dang, Truong & Huynh (2021)	Fuzzy clustering
algorithms for human resource grouping	No	No	
Huang & Wang (2022)	A human-resource management system for clustering employee talents	No	No	
Prasad, Choudhary & Ankayarkanni (2020)	An elbow-based K-means model for employee performance evaluation	No	No	
Arora & Badal (2013)	A K-means-based system for student performance evaluation	No	No	
Borgavakar & Shrivastava (2017)	K-means models for student performance evaluation using the sets of various attributes	No	No	
Parveen, Alphones & Naz (2017)	
Shankar et al. (2016)	
Xi (2015)	
Iqbal et al. (2019)	A Restricted-Boltzmann-Machine model for predicting student grades	No	No	
Ramen & Joachims (2014)	Probabilistic models for student peer grading	No	No	
Omar, Alzahrani & Zohdy (2020)	Elbow-based K-means models for student performance evaluation	No	No	
Pamungkas, Dewi & Putri (2024)	
Banditwattanawong & Masdisornchote (2020a)	Heuristic approaches for the norm-referenced achievement grading of learners	No	No	
Banditwattanawong & Masdisornchote (2020b)	
Banditwattanawong & Masdisornchote (2021)	
Banditwattanawong, Jankasem & Masdisornchote (2023)	A hybrid machine-learning and heuristic method for norm-referenced achievement grading	Yes	No	

Definition of conditional performance discretization

It is crucial to establish a rigorous foundation for proposing CPD algorithms and evaluating their performance. This section formally defines CPD that is applicable to both norm-referenced and criterion-referenced performance discretization schemes. Let P be a set of one-dimensional performance values, L be a set of assignable performance-ranking labels, La be a set of performance-ranking labels assigned to P, and fCPD() denote a CPD function that belongs to either norm-referenced or criterion-referenced scheme, CPD implements Eq. (1). Note that La ⊂ L indicates that La is a proper subset of L.

l∈La←fCPD(p∈P)

(1) suchthatLa⊂L.

This definition will be empirically exemplified through the following section.

Conditional unbiasedness requirements and metric for norm-referenced CPD

Intuitively, the norm-referenced CPD of descendingly-sorted performance values will achieve exact unbiasedness if three following requirements are completely fulfilled. Let PRL represent a linguistic performance-ranking label and PVI denote a performance-value interval, defined as a difference between the maximum and minimum performance values associated with the same PRL. Requirement 1: Every gap between adjacent performance values associated with two unique PRLs, including a gap between the top performance value and its upper bound, as well as a gap between the worst performance value and its lower bound that exceeds the largest PVI preserves a certain unique PRL from being assigned. This realizes the CPD property specified by Eq. (1).

Requirement 2: Every gap between contiguous performance-value boundaries associated with two unique PRLs is maximized. This prevents situations where any two closely situated performance values receive disparate PRLs, which may signify bias in a CPD process.

Requirement 3: All PVIs are uniform. This ensures that all assigned PRLs have the comparable chances of being assigned.

Conditional unbiasedness is achieved when the CPD process fairly assigns PRLs without favoring specific intervals of performance values. This is mathematically formalized through Eq. (2). These three requirements ensure conditional unbiasedness in Norm-Referenced CPD by preventing artificial PRL assignment in sparse performance regions (Requirement 1), ensuring that performance value gaps between PRLs do not create unfair distinctions (Requirement 2), and distributing PRLs evenly across performance intervals (Requirement 3). Requirements 2 and 3 are also common for UPD and were demonstrated in Banditwattanawong, Jankasem & Masdisornchote (2023).

We further clarify Requirement 1 via a simplified example presented in Table 2. In this example, a vector of 14 performance values is mapped to a vector of PRLs <A, B, C, D, E, F, G, H, I> by using a possible norm-referenced CPD method (A0). At an initial execution stage of A0, the results contain the same number of PVIs as the unique PRLs and include the gaps (four instances of 14 in bold) larger than the largest PVI of 5. This indicates that Requirement 1 has not yet been satisfied, thus requiring further adjustment of PRLs. To fulfill Requirement 1, specific unique PRLs must be sacrificed for these gaps by being excluded from assignment. In the final execution stage, A0 not only excludes A from being assigned due to the first gap of 14 but also reduces the number of PVIs by four causing the performance values 96, 92, 86, and 82 to be embraced into the same PVI associated with PRL B. Subsequently, the second gap of 14 between 82 and 77 excludes the associated PRL C from being assigned, while 77, 63, and 58 are grouped into the same PVI associated with a following PRL D. Likewise, the third gap of 14, following E, excludes F from assignment, and the last gap of 14, following H, excludes I from assignment. Consequently, no gap in the final stage exceeds the updated largest PVI of 19, thereby satisfying Requirement 1.

Table 2 Demonstration of requirement one satisfaction.

Bolding indicates the gaps (four instances of 14).

Performance value	Gap from previous performance value	Initial stage	Final stage	
		Initial PRL assigned by A0	Gap between unique PRLs and both bounds	PVI	Final PRL assigned by A0	Gap between unique PRLs and both bounds	PVI	
Upper bound	110	–	–	–	–	–	–	–	
Sorted values	96	14	A	14	4	B	14	14	
92	4	A	–	B	–	
86	6	B	6	4	B	–	
82	4	B	–	B	–	
77	14	C	14	0	D	14	19	
63	6	D	6	5	D	–	
58	4	D	–	D	–	
52	6	E	6	4	E	6	4	
48	4	E	–	E	–	
43	14	F	14	0	G	14	16	
27	6	G	6	0	G	–	
20	7	H	7	0	H	7	6	
15	5	I	5	1	H	–	
14	1	I	–	H	–	
Lower bound	0	14	–	14	–	–	14	–	

When applying norm-referenced CPD to real-world human performances, each of these requirements is often partially met. Hence, quantifying the attained degree of conditional unbiasedness in the deployment of norm-referenced CPD becomes beneficial for identifying the optimal CPD approach. Conditional unbiasedness metric Ω′ that measures the extent of conditional unbiasedness exhibiting in the outputs of a norm-referenced CPD approach can be figured out via Eq. (2). Ω′ is derived as the product of metrics, Ω1, Ω2, and Ω3, measuring the degrees of accomplishment for the Requirements 1 to 3, respectively. A higher Ω′ value indicates that the CPD process has minimized potential biases, thereby ensuring a robust and equitable evaluation system.

(2) Ω′=Ω1×Ω2×Ω3Ω1={1−|Θ−θ|ΘifΘ≥11,otherwiseΩ2={∑i=1N−1⁡δi−∑i=1N−1⁡Δi(min)∑i=1N−1⁡Δi(max)−∑i=1N−1⁡Δi(min)ifN≥31,otherwiseΩ3={11+σifN≥21,otherwise

where N denotes the number of unique PRLs assigned to performance values, δi denotes a gap between the lower-bound performance-value of the ith PRL and the upper-bound performance-value of the consecutive (i+1)th PRL (assuming performance values are sorted in descending order corresponding to that of the PRLs), Δi(min) denotes the ith narrowest gap between any adjacent performance values, Δi(max) denotes the ith widest gap between any adjacent performance values, σ denotes the standard deviation of the PVIs of all assigned unique PRLs, θ denotes the number of unassigned unique PRLs, and Θ denotes the number of both δi and a gap (γu) between the top performance value and its upper bound as well as a gap (γl) between the worst performance value and its lower bound that are larger than a PVI (denoted by νi(I)), which is widest and associated with a unique PRL. The value ranges of Ω1 is [0.0, 1.0], while Ω2, Ω3 have the range of (0.0, 1.0].

The rationale behind the calculation of Ω′, which is [0.0, 1.0], is as follows. Since Θ notates the count of PRLs that ought to remain unassigned, θ /Θ in Ω1 indicates the amount of CPD efficiency in satisfying Requirement 1 when assigning PRLs. If Ω1 equals 1.0, ∑i=1N−1⁡δi in Ω2 reaches its upper bound, ∑i=1N−1⁡Δi(max), (i.e., Requirement 2 is met), and σ in Ω3 is 0 (i.e., Requirement 3 is satisfied), Ω′ will be 1.0 indicating none of conditional biasedness. In contrast, as the term θ /Θ approaches 0, and/or ∑i=1N−1⁡δi approaches its lower bound, ∑i=1N−1⁡Δi(min), and/or σ approaches ∞, Ω′ will approach 0 indicating the low degree of conditional unbiasedness. Thus, a high value of Ω′ signifies a greater extent of conditional unbiasedness.

For clarification, Table 3 demonstrates the application of Eq. (2) through a vector of 30 descendingly-sorted performance values, a vector of assignable PRLs in a descendingly qualitative order <A, B, C, D, F>, and two possible norm-referenced CPD methods A1 and A2. Since A1 and A2 assign 4 and 3 PRLs, respectively, to different PVIs, the Ω′ values of CPD outputs delivered by A1 and A2 where N = 4 and 3, respectively, are calculated as follows:

A1:Ω′=(1−|2−1|2)×(10−0)(10−0)×1(1+5.07)=0.08A2:Ω′=(1−|2−2|2)×(6−0)(7−0)×1(1+2.89)=0.22.

Table 3 Demonstration of Ω′ calculation of two possible CPD algorithms.

Performance value	Gap from previous performance value	A1	A2	
		PRL assigned by A1	γu, δi, γl	νi(I)	PRL assigned by A2	γu, δi, γl	νi(I)	
Upper bound	100	–	–	–	–	–	–	–	
Sorted values	82	18	A	18	2	B	18	7	
80	2	A	–	B	–	
76	4	B	4	1	B	–	
75	1	B	–	B	–	
72	3	C	3	7	C	3	7	
70	2	C	–	C	–	
69	1	C	–	C	–	
69	0	C	–	C	–	
68	1	C	–	C	–	
68	0	C	–	C	–	
67	1	C	–	C	–	
65	2	C	–	C	–	
65	0	C	–	C	–	
62	3	D	3	12	D	3	12	
61	1	D	–	D	–	
59	1	D	–	D	–	
58	1	D	–	D	–	
57	1	D	–	D	–	
57	0	D	–	D	–	
57	0	D	–	D	–	
56	1	D	–	D	–	
56	0	D	–	D	–	
55	1	D	–	D	–	
54	1	D	–	D	–	
53	1	D	–	D	–	
52	1	D	–	D	–	
52	0	D	–	D	–	
51	1	D	–	D	–	
50	1	D	–	D	–	
50	0	D	–	D	–		
Lower bound	0	50	–	50	–	–	50	–	

The reason for A2 exhibiting a higher degree of conditional unbiasedness compared to A1 is that A2 more effectively satisfies Requirements 1 and 3.

Norm-referenced CPD methodology

This section proposes four unbiasedness-centric methods for norm-referenced CPD: widest-gap-first CPD method (WGF-CPD), K-means CPD method (K-CPD), Partitioning-Around-Medoids CPD method (PAM-CPD), and multi-modal CPD method (M-CPD).

Widest-gap-first CPD method

Starting with all performance values, this method employs a heuristic approach, namely widest-gap-first, to identify the largest gaps (i.e., the widest δi) for categorizing performance values when assigning PRLs. The method iteratively reduces the number of PRLs assigned to ensure compliance with CPD requirements. Performance values falling into distinct categories are assigned different PRLs. WGF-CPD is simple and interpretable compared to the other CPD methods.

The method is outlined in Algorithm 1. Line 1 calculates gaps between every pair of adjacent performance values and select only (the total number of unique PRLs – 1) widest gaps. If there are multiple identical widest gaps, the one leading to more uniform PVIs (minimizing σ in Eq. (2)) will be selected. In line 2, heuristicPD() discretizes performance values based on the widest-gap-first heuristic approach by initially assigning all unique PRLs to all performance values grouped according to the |l|−1 widest gaps. Line 3 determines PVIs associated with each unique PRL. Line 4 calculates gaps γu, γl, and δi=1 ..|l|−1. Line 5 counts the number of gaps from Line 4 that their gap sizes equal or exceed the largest PVI from Line 3. Line 6 sets the number of unassigned PRLs initially to zero. Line 7 calculates Eq. (2). Line 8 sets the positive-integer index j (representing the number of sacrificed or unassigned PRLs) to 1. Line 9 iterates lines 10 to 17 only if the condition j ≤ Θ holds true. Each iteration sacrifices a unique PRL for each of the widest gaps equaling or exceeding the largest PVI by incrementing j by 1 from 1 to Θ. Lines 10 to 17 execute similar operations to lines 1 to 8, but constrained to the |l|-1−j widest gaps. Each iteration implies that there is a gap in g wider than maxElement(ν(I)) causing heuristicPD() in line 11, which is the overloading variant of heuristicPD() in line 2, to sacrifice a respective PRL in l for some gap not only in g(max) but also γu and γl exceeding the widest PVI and totally assigns |l|−j PRLs in each iteration. (The process of excluding PRLs from assignment has been demonstrated through Table 2.) Notice that the iteration might terminate when Θ is still greater than 0 meaning that Requirement 1 fulfillment is only partial. Line 15 updates θ value to be used for Ω′ calculation in the next iteration. Line 17 increases j by 1. Line 18 identifies the index value of k for which Ωk=0..j−1′ is maximized. Line 19 returns the output vector of PRL instances, delivering the maximum Ωk′, that are aligned with each performance value in the input ν one by one. In fact, if Θ in line 5 equals 0, the while loop will be bypassed causing Algorithm 1 to function as a UPD method.

Algorithm 1 Widest-gap-first CPD algorithm.

Algorithm WGF-CPD	
Input	
  ν   : vector of performance values in descending order	
  U  : upper-bound performance value	
  L   : lower-bound performance value	
  l   : vector of unique PRLs in a descendingly qualitative order	
Output	
  l(o)  : vector of PRL instances that matches ν both size and order	
Local variable	
  g(max) : vector of widest performance-value gaps	
  g  : vector of full-range performance-value gaps <γu, δi=1..|l|-1, γl>	
  ν(I)  : vector of PVIs	
   lj(o)  : jth vector of PRL instances that matches ν in both size and order	
   Ωj′  : jth Ω′ value	
Begin	
 g(max) ← findWidestGaps(v, |l| - 1);

  lj=0(o)  ← heuristicPD(ν, l, g(max));

 ν(I)  ← getPVIs( lj=0(o), ν);

 g  ← getPerformanceValueGaps(ν, lj=0(o), U, L);

 Θ   ← |{m∈g | m ≥ maxElement(ν(I))}|;

 θ  ← 0;

  Ωj=0′  ← getConditionalUnbiasedness(ν, lj=0(o), U, L, θ, Θ);

 j ← 1;

 while j ≤ Θ

  g(max) ← findWidestGaps(ν, |l| – 1 – j);

   lj(o)  ← heuristicPD(ν, l, g(max), γu ∈ g, γl ∈ g);

  ν(I)  ← getPVIs( lj(o), ν);

  g  ← getPerformanceValueGaps(ν, lj(o), U, L);

  Θ   ← |{m∈g | m ≥ maxElement(ν(I))}|;

  θ  ← j;

   Ωj′  ← getConditionalUnbiasedness(ν, lj(o), U, L, θ, Θ);

  j   ← j+1;

 k  ← argmaxk( Ωk=0..j−1′);

 l(o)  ← lk(o);

	
End	

The practical cost-effectiveness of Algorithm 1 is analyzed in terms of worst-case time complexity as follows. Let n and N be the numbers of performance values (i.e., |v|) and unique PVLs (i.e., |l|), respectively. To sort input v and l takes n log2n and N log2N, respectively. Line 1 takes (N-1) log2n. Line 2 takes n. Lines 3–5 identically take n. Line 7 is 3N+n log2n to figure out all parameters and calculate Eq. (1). Line 9 iterates at most N+1 rounds (as Θ is bounded by |g|, which is at most N+1) of lines 10–17, which are equivalent to lines 2–8. Line18 takes N+1. Line 19 takes n. Therefore, WGF-CPD algorithm is O(n log2n + N log2N + (N−1) log2n + n + 3n + 3N+n log2n + (N+1)((N−1) log2n + n + 3n + 3N + n log2n) + N+1+n) that is O(n log2n) as N << n.

K-means CPD method

This method engages K-means (Witten & Frank, 2016) for partitioning performance values into a predefined number, K, of clusters, each represents a unique PRL. Unlike supervised methods, this approach does not use labeled data but instead minimizes the variance within clusters to optimize grouping. In the context of CPD, the number of clusters is fewer than the number of assignable unique PRLs. The method iteratively adjusts clusters to align with performance gaps and ensures compliance with the CPD requirements, particularly focusing on equalizing PVIs while maintaining the widest-gap constraints. This ensures unbiased distribution, as measured by Ω′.

In specific, K-means method minimizes an objective function ∑j=1K⁡∑i=1nj⁡|xi−cj| where nj is the number of performance values in cluster j, xi is a performance value in j with a centroid cj, and |xi − cj| is Euclidean distance. K-CPD method invokes K-means more than once to produce CPD results and the best one with the highest Ω′ is finally returned as showed in Algorithm 2. Several lines of code are the identical to those in Algorithm 1 and have been explained with the exception of the followings. In line 1, KmeansPD() performs performance discretization by assigning all unique PRLs to all performance values grouped by using K-means method into |l| clusters. During each iteration, the overloading function KmeansPD() will not assign a unique PRL corresponding to a performance value gap between U and L equal to or wider than maxElement(ν(I)). As long as the condition j ≤ Θ in line 8 is not evaluated to false, the algorithm continues to invoke K-means algorithm to re-cluster the performance values into |l|−j clusters. In each iteration of the while-loop, the resulting PRL-instance vector l contains a unique set of PRLs or clusters that differ from those in other iterations and is assumed to be stored. Finally, in lines 16–17, the K-CPD algorithm returns the vector of PRL instances that achieves the highest conditional unbiasedness.

Algorithm 2 K-means CPD algorithm.

Algorithm K-CPD	
Input	
  ν   : vector of performance values in descending order	
  U  : upper-bound performance value	
  L  : lower-bound performance value	
  l   : vector of unique PRLs in a descendingly qualitative order	
Output	
  l(o)  : vector of PRL instances that matches ν both size and order	
Local variable	
  g   : vector of full-range performance-value gaps <γu, δi=1..|l|-1, γl>	
  ν(I)  : vector of PVIs	
   lj(o)  : jth vector of PRL instances that matches ν in both size and order	
   Ωj′  : jth Ω′ value	
Begin	
  lj=0(o)  ← KmeansPD(ν, l, |l|);

 ν(I) ← getPVIs( lj=0(o), ν);

 g   ← getPerformanceValueGaps(ν, lj=0(o), U, L);

 Θ   ← |{m∈g | m ≥ maxElement(ν(I))}|;

 θ   ← 0;

  Ωj=0′ ← getConditionalUnbiasedness(ν, lj=0(o), U, L, θ, Θ);

 j   ← 1;

while j ≤ Θ

   lj(o) ← KmeansPD(ν, l, |l|−j, U, L);

  ν(I) ← getPVIs( lj(o), ν);

  g ← getPerformanceValueGaps(ν, lj(o), U, L);

  Θ  ← |{m∈g | m ≥ maxElement(ν(I))}|;

  θ ← j;

   Ωj′ ← getConditionalUnbiasedness(ν, lj(o), U, L, θ, Θ);

  j  ← j+1;

 k  ← argmaxk( Ωk=0..j−1′);

 l(o) ← lk(o);

	
End	

The time complexity of Algorithm 2 is analyzed as follows. To sort input v and l takes n log2n and N log2N, respectively. Line 1 takes cNn where N is equivalent to the number of clusters and hyper parameter c is the bound number of internal iterations in each complete K-means execution. Lines 2–4 identically take n. Line 6 is 3N+n log2n. Line 8 iterates at most N+1 rounds of lines 9–15, which are equivalent to lines 1–7. Line 16 takes N+1. Line 17 takes n. Therefore, K-CPD algorithm is O(n log2n) as N << n. However, for large data sets, K-CPD becomes sensitive to the initial placement of centroids, potentially leading to inconsistent results across different runs. This issue can be mitigated by replacing random initialization with more reliable techniques, such as K-Means++.

Partitioning-around-medoids CPD method

This method adapts K-CPD by replacing K-means with PAM (Kaufmann & Rousseeuw, 1987), which enhances robustness to outliers. PAM is different from K-means in how it selects cluster centers, as PAM choose actual data points (medoids) as cluster centers instead of means. This makes PAM more robust to outliers, whereas K-means is sensitive to extreme values that can distort cluster assignments. However, this robustness comes at a computational cost, as PAM has a higher complexity of O(n2N), making it slower than K-means, which operates at O(cNn). Unlike K-means, which can produce different results due to random initialization, PAM provides more stable clustering outcomes. Thus, PAM is preferred for small datasets, while K-means is better suited for large-scale clustering tasks requiring efficiency.

Algorithm 3 details PAM-CPD method. The algorithm differs from Algorithm 2 in lines 1 and 9 substituting PAM_PD() for KmeanPD(). PAM_PD() discretizes the performance vector v by employing PAM method into |l| and |l|−j PRLs or clusters. Each while loop generates at most Θ PVL vectors. The algorithm returns a PVL vector with the maximum Ω′.

Algorithm 3 PAM CPD algorithm.

Algorithm PAM-CPD	
Input	
  ν  : vector of performance values in descending order	
  U  : upper-bound performance value	
  L  : lower-bound performance value	
  l  : vector of unique PRLs in a descendingly qualitative order	
Output	
  l(o) : vector of PRL instances that matches ν in both size and order	
Local variable	
  g  : vector of full-range performance-value gaps <γu, δi=1..|l|-1, γl>	
  ν(I) : vector of PVIs	
   lj(o) : jth vector of PRL instances that matches ν both size and element order	
   Ωj′ : jth Ω′ value	
Begin	
  lj=0(o)  ← PAM_PD(ν, l, |l|);

 ν(I)  ← getPVIs( lj=0(o), ν);

 g   ← getPerformanceValueGaps(ν, lj=0(o), U, L);

 Θ  ← |{m∈g | m ≥ maxElement(ν(I))}|;

 θ   ← 0;

  Ωj=0′ ← getConditionalUnbiasedness(ν, lj=0(o), U, L, θ, Θ);

 j   ← 1;

 while j ≤ Θ

   lj(o) ← PAM_PD(ν, l, |l|−j, U, L);

  ν(I) ← getPVIs( lj(o), ν);

  g ← getPerformanceValueGaps(ν, lj(o), U, L);

  Θ  ← |{m∈g | m ≥ maxElement(ν(I))}|;

  θ  ← j;

   Ωj′ ← getConditionalUnbiasedness(ν, lj(o), U, L, θ, Θ);

  j ← j+1;

 k  ← argmaxk( Ωk=0..j−1′);

 l(o) ← lk(o);

	
End	

The time complexity of Algorithm 3 can be derived by replacing cNn with n2N. This makes PAM-CPD, which is O(n2), still tractable.

Multi-modal CPD method

M-CPD method leverages both WGF-CPD, K-CPD, and PAM-CPD methods to process the same set of inputs and return the results of either constituent method with the highest Ω′. M-CPD method takes the number of assignable unique PRLs to specify not only the initial number of clusters determined by K-CPD and PAM-CPD methods but also the initial number of PVIs into which WGF-CPD method divides performance values. M-CPD method employs conditional-unbiasedness maximization as its decision function, which is grounded in Ω′ to quantify the degree to which a norm-referenced CPD process satisfies the three core requirements justified below. Particularly, the decision function evaluates the results of WGF-CPD, K-CPD, and PAM-CPD, and selects the one with the highest Ω′, ensuring optimal conditional unbiasedness. Requirement 1: Unassigned PRLs are maximized for gaps larger than or equal to the widest PVI, ensuring no bias arises from arbitrary assignments in sparsely populated regions.

Requirement 2: Gaps between assigned PRLs are maximized, preventing closely spaced performance values from receiving different PRLs, which could introduce unfair distinctions.

Requirement 3: Uniformity of PVIs is maintained to equalize the likelihood of PRLs being assigned across the performance spectrum.

Algorithm 4 provides the details of M-CPD method. Lines 1 to 3 call WGF-CPD, K-CPD, and PAM-CPD algorithms, respectively, to process input performance values and PRLs. In line 4, the algorithm implements the maximization of its decision function to identify the best CPD result among a pair generated by the constituent algorithms. Note that getConditionalUnbiasedness() in line 4 overloads the function getConditionalUnbiasedness() in Algorithms 1, 2, and 3 by calculating θ and Θ merely from the input parameters ν, lj(o), U, and L.

Algorithm 4 Unbiasedness-centric multi-modal CPD algorithm.

Algorithm M-CPD	
Input	
  ν   : vector of performance values in descending order	
  U   : upper-bound performance value	
  L   : lower-bound performance value	
  l     : vector of unique PRLs in a descendingly qualitative order	
Output	
  l(o)  : vector of PRL instances that matches ν in both size and order	
Begin	
  l1(o) ← WGF-CPD(ν, l, U, L);

  l2(o) ← K-CPD(ν, l, U, L);

  l3(o) ← PAM-CPD(ν, l, U, L);

 l(o) ← argmaxlj(o)∈{l1(o),l2(o),l3(o)} getConditionalUnbiasedness(ν, lj(o), U, L);

	
End	

The worst-case time complexity of Algorithm 4 is as follows. To sort input v and l takes n log2n and N log2N, respectively. Lines 1 and 2 equally takes n log2n. Line 3 takes n2. Line 4 is 2(3N+n log2n). Therefore, M-CPD algorithm totally takes 3n log2n + N log2N + n2 + 2(3N+n log2n) equaling O(n2).

Evaluation

The proposed methods are assessed by using open data sets and analyzed in terms of conditional unbiasedness as outlined below.

Data set and preprocessing

Multiple data sets from human-performance evaluation domains, each of which consists of one or more fields, are preprocessed to derive stakeholder-interpretable and one-dimentional performance values as follows. The statistical characteristics of each data set are described via a vector of <maximum performance value, minimum performance value, performance value mean, standard deviation, distribution skewness>, providing a comprehensive summary of the data’s central tendency, variability, and shape.

The first data set, entitled EMP1, concerns employee evaluation for promotion consideration. To derive EMP1, the following fields were selected from the raw data set (Zaman, 2022) and max–min normalized into a range of 0.00 and 1.00: previous year rating (between 1 and 5), length of service in years, won awards (1 for won, 0 otherwise), number of trainings (ranging from 1 to 10), and average training score (between 0 and 100). Subsequently, the normalized values were summed to obtain individual performance values ranging from 0.00 to 5.00. The sampled number of employees evaluated is 100. These 100 preprocessed performance values will be then discretized conditionally into three promotional levels as suggested by the raw data set’s meta data: executive (denoted by A), higher-qualified professional (B), and non-qualified promotion (C). The statistical characteristics of this data set are <3.00, 0.79, 1.97, 0.46, −0.84>.

The second data set, namely EMP2, pertains to employee performance rating correlated with salary increases. To obtain EMP2, the raw data set (Kumar, 2022) including the field of percentage salary hikes was filtered in randomly only 60 records of employees. The preprocessed data will be discretized conditionally into performance ratings ranging from 1 to 4. A lower rating corresponds to a greater salary increase. The statistical characteristics of this data set are <24, 11, 15.66, 3.64, 0.25>.

The third data set, named STD1, involves the evaluation of performance among higher education students. In the raw data set (Yilmaz & Sekeroglu, 2022), a field representing the 1 to 5 ratings of cumulative grade point average (CGPA): 1 for < 2.00 CGPA, 2 for 2.00–2.49 CGPA, 3 for 2.50–2.99 CGPA, 4 for 3.00–3.49 CGPA, and 5 for above 3.49 CGPA. This field was sampled into 50 records and transformed from interval ratings to random CGPAs by applying randomization. These preprocessed CGPA values are to be discretized conditionally into grade symbols: AA, AB, BB, BC, CC, CD, DD, and Fail. The statistical characteristics of STD1 are <3.46, 0.18, 2.45, 0.86, −1.00>.

The final dataset, STD2, represents student performance in exams. It was derived by weighting math, reading, and writing scores from the raw dataset (Chauhan, 2023), where each subject has a full score of 100. A subset of 40 samples was selected for conditional discretization into five grade symbols (A, B, C, D, and F). The statistical characteristics of STD2 are <90, 23, 53.28, 22.30, 0.17>.

Experimental setup

The experiments were conducted in the following steps to evaluate the proposed norm-referenced CPD methods, WGF-CPD, K-CPD, PAM-CPD, and M-CPD. First, each preprocessed dataset was conditionally discretized separately using all proposed methods. The algorithmic processing is also concisely clarified. Subsequently, Ω′ value was computed for each combination of method and data set to quantify the conditional-unbiasedness performance of each method. Finally, the results are evaluated by comparison across the methods.

CPD result

The CPD results for each proposed method are presented via each complete data set in Table 4 onward. The conditional unbiasedness degrees are measured to observe the contributions of the constituent methods to the final results delivered by M-CPD method.

Table 4 CPD results of EMP1 data set.

Performance value (sorted)	Gap from either preceding performance value or upper bound	WGF-CPD	K-CPD	PAM-CPD	Z score	
Assigned PRL	γu, δi, γl	νi(I)	Assigned PRL	γu, δi, γl	νi(I)	Assigned PRL	γu, δi, γl	νi(I)	Assigned PRL	γu, δi, γl	νi(I)	
5.00 (upper bound)	–	–	–	–	–	–	–	–	–	–	–	–	–	
3.00	2.00	B	2.00	0.00	B	2.00	1.23	B	2.00	2.09	A	2.00	0.73	
2.80	0.20	C	0.20	2.01	B	–	B	–	A	–	
2.69	0.01	C	–	B	–	B	–	A	–	
2.63	0.06	C	–	B	–	B	–	A	–	
2.62	0.01	C	–	B	–	B	–	A	–	
2.57	0.05	C	–	B	–	B	–	A	–	
2.50	0.07	C	–	B	–	B	–	A	–	
2.49	0.01	C	–	B	–	B	–	A	–	
2.46	0.03	C	–	B	–	B	–	A	–	
2.44	0.02	C	–	B	–	B	–	A	–	
2.42	0.02	C	–	B	–	B	–	A	–	
2.40	0.02	C	–	B	–	B	–	A	–	
2.40	0.00	C	–	B	–	B	–	A	–	
2.37	0.03	C	–	B	–	B	–	A	–	
2.36	0.01	C	–	B	–	B	–	A	–	
2.36	0.00	C	–	B	–	B	–	A	–	
2.36	0.00	C	–	B	–	B	–	A	–	
2.36	0.00	C	–	B	–	B	–	A	–	
2.36	0.00	C	–	B	–	B	–	A	–	
2.36	0.00	C	–	B	–	B	–	A	–	
2.35	0.01	C	–	B	–	B	–	A	–	
2.34	0.01	C	–	B	–	B	–	A	–	
2.32	0.02	C	–	B	–	B	–	A	–	
2.31	0.01	C	–	B	–	B	–	A	–	
2.31	0.00	C	–	B	–	B	–	A	–	
2.27	0.04	C	–	B	–	B	–	A	–	
2.26	0.01	C	–	B	–	B	–	B	0.01	0.72	
2.26	0.00	C	–	B	–	B	–	B	–	
2.26	0.00	C	–	B	–	B	–	B	–	
2.25	0.01	C	–	B	–	B	–	B	–	
2.25	0.00	C	–	B	–	B	–	B	–	
2.24	0.01	C	–	B	–	B	–	B	–	
2.23	0.01	C	–	B	–	B	–	B	–	
2.22	0.01	C	–	B	–	B	–	B	–	
2.20	0.02	C	–	B	–	B	–	B	–	
2.19	0.01	C	–	B	–	B	–	B	–	
2.17	0.02	C	–	B	–	B	–	B	–	
2.15	0.02	C	–	B	–	B	–	B	–	
2.13	0.02	C	–	B	–	B	–	B	–	
2.13	0.00	C	–	B	–	B	–	B	–	
2.12	0.01	C	–		B	–		B	–		B	–		
2.11	0.01	C	–	B	–	B	–	B	–	
2.10	0.01	C	–	B	–	B	–	B	–	
2.10	0.00	C	–	B	–	B	–	B	–	
2.10	0.00	C	–	B	–	B	–	B	–	
2.09	0.01	C	–	B	–	B	–	B	–	
2.08	0.01	C	–	B	–	B	–	B	–	
2.07	0.01	C	–	B	–	B	–	B	–	
2.07	0.00	C	–	B	–	B	–	B	–	
2.07	0.00	C	–	B	–	B	–	B	–	
2.07	0.00	C	–	B	–	B	–	B	–	
2.06	0.01	C	–	B	–	B	–	B	–	
2.06	0.00	C	–	B	–	B	–	B	–	
2.06	0.00	C	–	B	–	B	–	B	–	
2.05	0.01	C	–	B	–	B	–	B	–	
2.04	0.01	C	–	B	–	B	–	B	–	
2.04	0.00	C	–	B	–	B	–	B	–	
2.03	0.01	C	–	B	–	B	–	B	–	
2.02	0.01	C	–	B	–	B	–	B	–	
2.01	0.01	C	–	B	–	B	–	B	–	
2.01	0.00	C	–	B	–	B	–	B	–	
2.01	0.00	C	–	B	–	B	–	B	–	
2.00	0.01	C	–	B	–	B	–	B	–	
2.00	0.00	C	–	B	–	B	–	B	–	
1.99	0.01	C	–	B	–	B	–	B	–	
1.98	0.01	C	–	B	–	B	–	B	–	
1.94	0.04	C	–	B	–	B	–	B	–	
1.94	0.00	C	–	B	–	B	–	B	–	
1.93	0.01	C	–	B	–	B	–	B	–	
1.92	0.01	C	–	B	–	B	–	B	–	
1.82	0.10	C	–	B	–	B	–	B	–	
1.78	0.04	C	–	B	–	B	–	B	–	
1.77	0.01	C	–	B	–	B	–	B	–	
1.70	0.07	C	–	C	0.07	0.91	B	–	B	–	
1.65	0.05	C	–	C	–	B	–	B	–	
1.62	0.03	C	–	C	–	B	–	B	–	
1.61	0.01	C	–	C	–	B	–	B	–	
1.54	0.07	C	–	C	–	B	–	B	–	
1.54	0.00	C	–	C	–	B	–	B	–	
1.54	0.00	C	–	C	–	B	–	B	–	
1.54	0.00	C	–	C	–	B	–	B	–	
1.51	0.03	C	–		C	–		B	–		C	0.03	0.72	
1.45	0.06	C	–	C	–	B	–	C	–	
1.44	0.01	C	–	C	–	B	–	C	–	
1.35	0.09	C	–	C	–	B	–	C	–	
1.34	0.01	C	–	C	–	B	–	C	–	
1.34	0.00	C	–	C	–	B	–	C	–	
1.33	0.01	C	–	C	–	B	–	C	–	
1.31	0.02	C	–	C	–	B	–	C	–	
1.30	0.01	C	–	C	–	B	–	C	–	
1.21	0.09	C	–	C	–	B	–	C	–	
1.15	0.06	C	–	C	–	B	–	C	–	
1.13	0.02	C	–	C	–	B	–	C	–	
1.13	0.00	C	–	C	–	B	–	C	–	
1.11	0.02	C	–	C	–	B	–	C	–	
1.10	0.01	C	–	C	–	B	–	C	–	
1.04	0.06	C	–	C	–	B	–	C	–	
0.94	0.10	C	–	C	–	B	–	C	–	
0.91	0.03	C	–	C	–	B	–	C	–	
0.79	0.12	C	–	C	–	C	0.12	0.00	C	–	
0.00
(lower bound)	0.79	–	0.79	–	–	0.79	–	–	0.79	–	–	0.79	–	

In Table 4, EMP1 data set is processed independently by WGF-CPD, K-CPD, and PAM-CPD methods as well as baseline Z-score method as follows.

Initially, WGF-CPD method discretizes the performance values into three PRLs A, B, and C, each of which is associated with the performance values [3.00], [2.80–0.91], and [0.79], respectively, resulting in PVIs of <0.00, 1.89, 0.00> (not showed in Table 4). However, there remains the gap γu of 2.00 wider than the maximum PVI (1.89) and preceding the maximum performance value, thus PRL A should have been sacrificed. As a result, WGF-CPD re-discretizes the performance values into only 2 PRLs, B and C, as showed in Table 4. Notably, no gap wider than the maximum PVI (2.01) remains, leading to the completion of the method.

K-CPD method begins by grouping the performance values into 3 clusters: [3.00, 2.22], [2.20, 1.70], and [1.65, 0.79], associated with PRLs A, B, and C, respectively (not shown in Table 4). Since all PVIs <0.78, 0.50, 0.86> are smaller than γu, K-CPD re-groups the performance values into 2 clusters, [3.00, 1.77] and [1.70, 0.79], associated with B and C, respectively, as demonstrated in Table 4. However, the widest PVI of 1.23 still remains smaller than γu although K-CPD terminates since the condition in line 8 of Algorithm 2 evaluates to false (i.e., the number of excluded PRLs reached Θ).

PAM-CPD method initially groups the performance values into 3 clusters: [3.00, 1.04], [0.94, 0.91], and [0.79, 0.79], associated with PRLs A, B, and C, respectively, (not showed in Table 4). Since all PVIs <1.96, 0.03, 0.00> are smaller than γu, PAM-CPD re-groups the performance values into two clusters, [3.00, 0.91] and [0.79, 0.79], associated with B and C, respectively, as demonstrated in Table 4. The widest PVI of 2.09 is larger than γu, causing Θ to become 0 and terminating the while loop in Algorithm 3.

The CPD results in Table 4 produced by WGF-CPD, K-CPD, PAM-CPD, and Z-score methods exhibit different degrees of Ω′ as calculated below based on Eq. (2) with N = 2. Greater Ω′ value indicates that K-CPD offers higher degree of conditional unbiasedness. Consequently, M-CPD method returns the CPD result produced by constituent K-CPD. The baseline method, Z-score, fully assigns PVLs although γu is greater than 0.73, resulting in zero Ω1, which leads to a failure in performing CPD for this data set.

WGF-CPD:Ω′=1×1×11+1.42=0.41K-CPD:Ω′=(1−01)×1×11+0.23=0.82PAM-CPD:Ω′=1×1×11+1.48=0.40Z-score:Ω′=0×0.04−0.000.32−0.00×11+0.01=0.00.

By applying WGF-CPD, EMP2 data set is conditionally discretized into PRLs 1 to 4, corresponding to performance value ranges [24, 24], [22, 17], [15, 13], and [11, 11] with PVIs 0, 5, 2, and 0, respectively. However, the gap γl equals 6, larger than the maximum PVI of 5 resulting in WGF-CPD method reallocating PRLs 1, 2, and 3 to performance value ranges [24, 17], [15, 13], and [11, 11] with adjusted PVIs 7, 2, and 0, respectively, as illustrated in Table 5. This ensures that γl does not exceed the maximum PVI of 7, thus completing WGF-CPD operation.

Table 5 CPD results of EMP2 data set.

Performance value (sorted)	Gap from either preceding performance value or upper bound	WGF-CPD	K-CPD	PAM-CPD	Z score	
Assigned PRL	γu, δi, γl	νi(I)	Assigned PRL	γu, δi, γl	νi(I)	Assigned PRL	γu, δi, γl	νi(I)	Assigned PRL	γu, δi, γl	νi(I)	
25 (upper bound)	–	–	–	–	–	–	–	–	–	–	–	–	–	
24	1	1	1	7	1	1	4	1	1	7	1	1	3	
24	0	1	–	1	–	1	–	1	–	
22	2	1	–	1	–	1	–	1	–	
22	0	1	–	1	–	1	–	1	–	
21	1	1	–	1	–	1	–	1	–	
21	0	1	–	1	–	1	–	1	–	
21	0	1	–	1	–	1	–	1	–	
20	1	1	–	1	–	1	–	2	1	2	
20	0	1	–	1	–	1	–	2	–	
20	0	1	–	1	–	1	–	2	–	
20	0	1	–	1	–	1	–	2	–	
19	1	1	–	2	1	2	1	–	2	–	
19	0	1	–	2	–	1	–	2	–	
19	0	1	–	2	–	1	–	2	–	
19	0	1	–	2	–	1	–	2	–	
19	0	1	–	2	–	1	–	2	–	
18	1	1	–	2	–	1	–	2	–	
18	0	1	–	2	–	1	–	2	–	
18	0	1	–	2	–	1	–	2	–	
18	0	1	–	2	–	1	–	2	–	
18	0	1	–	2	–	1	–	2	–	
18	0	1	–	2	–	1	–	2	–	
18	0	1	–	2	–	1	–	2	–	
17	1	1	–	2	–	1	–	3	1	2	
17	0	1	–	2	–	1	–	3	–	
17	0	1	–	2	–	1	–	3	–	
17	0	1	–	2	–	1	–	3	–	
17	0	1	–	2	–	1	–	3	–	
15	2	2	2	2	3	2	4	2	2	2	3	–	
15	0	2	–	3	–	2	–	3	–	
15	0	2	–	3	–	2	–	3	–	
15	0	2	–	3	–	2	–	3	–	
15	0	2	–	3	–	2	–	3	–	
15	0	2	–	3	–	2	–	3	–	
14	1	2	–	3	–	2	–	4	1	3	
14	0	2	–	3	–	2	–	4	–	
14	0	2	–	3	–	2	–	4	–	
14	0	2	–	3	–	2	–	4	–	
14	0	2	–		3	–		2	–		4	–		
13	1	2	–	3	–	2	–	4	–	
13	0	2	–	3	–	2	–	4	–	
13	0	2	–	3	–	2	–	4	–	
13	0	2	–	3	–	2	–	4	–	
13	0	2	–	3	–	2	–	4	–	
13	0	2	–	3	–	2	–	4	–	
13	0	2	–	3	–	2	–	4	–	
13	0	2	–	3	–	2	–	4	–	
11	2	3	2	0	3	–	3	2	0	4	–	
11	0	3	–	3	–	3	–	4	–	
11	0	3	–	3	–	3	–	4	–	
11	0	3	–	3	–	3	–	4	–	
11	0	3	–	3	–	3	–	4	–	
11	0	3	–	3	–	3	–	4	–	
11	1	3	–	3	–	3	–	4	–	
11	0	3	–	3	–	3	–	4	–	
11	0	3	–	3	–	3	–	4	–	
11	0	3	–	3	–	3	–	4	–	
11	0	3	–	3	–	3	–	4	–	
11	0	3	–	3	–	3	–	4	–	
11	0	3	–	3	–	3	–	4	–	
5 (lower bound)	6	–	6	–	–	6	–	–	6	–	–	6	–	

K-CPD method initially assigns PVLs 1 to 4 to the performance value ranges [24, 20], [19, 17], [15, 13], and [11, 11] with PVIs of 4, 2, 2, 0, respectively. Consequently, the maximum PVI of 4 smaller than γl causes K-CPD to consolidate the performance values into three clusters [24, 20], [19, 17], and [15, 11] associated with PVLs 1, 2, and 3, respectively. Despite this adjustment, γl still exceeds the maximum PVI of 4 whereas K-CPD terminates.

PAM-CPD method begins by assigning PVLs 1 to 4 to the performance value ranges [24, 18], [17, 17], [15, 13], and [11, 11] with PVIs of 6, 0, 2, 0, respectively. Consequently, the maximum PVI of 6 equaling γl causes PAM-CPD to consolidate the performance values into three clusters [24, 17], [15, 13], and [11, 11] associated with PVLs 1, 2, and 3, respectively. Consequently, γl is smaller than the maximum PVI of 7 and PAM-CPD terminates.

The CPD outputs in Table 5 where WGF-CPD and PAM-CPD methods deliver the identical results possess the following Ω′, indicating that K-CPD yields higher degree of conditional unbiasedness. Consequently, M-CPD method selects the CPD result of K-CPD. Z-score method assigns all PVLs although γl is greater than 3, resulting in zero Ω1.

WGF-CPD:Ω′=1×4−04−0×11+3.606=0.217K-CPD:Ω′=(1−01)×3−04−0×11+1.155=0.348PAM-CPD:Ω′=1×4−04−0×11+3.606=0.217Z-score:Ω′=0×3−06−0×11+0.577=0.000

Initially, STD1 data set is discretized by WGF-CPD method into eight PVLs representing performance value ranges [3.46, 3.04], [2.88, 2.50], [2.36, 2.00], [1.78, 1.71], [1.55, 1.55], [1.33, 1.27], [0.93, 0.93], and [0.37, 0.18] with the largest PVI of 0.42, which is smaller than the widest gap of 0.56. Consequently, WGF-CPD excludes AA from PVLs resulting in an adjusted CPD result as showed in Table 6 along with the largest PVI of 0.88 larger than the widest gap of 0.56. In contrast, K-CPD method begins by discretizing STD1 into eight PVLs with the largest PVI of 0.40, less than γu. Even after K-CPD unassigns AA, the newly largest PVI (0.45) still remains smaller than the widest gap of 0.56 as depicted in Table 6. PAM-CPD discretizes STD1 into eight PVLs with the largest PVI of 1.35, greater than γu. Therefore, PAM-CPD does not exclude any PVL from assignment at all.

Table 6 CPD results of STD1 data set.

Performance value (sorted)	Gap from either preceding performance value or upper bound	WGF-CPD	K-CPD	PAM-CPD	Z score	
Assigned PRL	γu, δi, γl	νi(I)	Assigned PRL	γu, δi, γl	νi(I)	Assigned PRL	γu, δi, γl	νi(I)	Assigned PRL	γu, δi, γl	νi(I)	
4.00 (upper bound)	–	–	–	–	–	–	–	–	–	–	–	–	–	
3.46	0.54	AB	0.54	0.42	AB	0.54	0.34	AA	0.54	1.35	AA	0.54	0.34	
3.46	0.00	AB	–	AB	–	AA	–	AA	–	
3.45	0.01	AB	–	AB	–	AA	–	AA	–	
3.42	0.03	AB	–	AB	–	AA	–	AA	–	
3.42	0.00	AB	–	AB	–	AA	–	AA	–	
3.36	0.06	AB	–	AB	–	AA	–	AA	–	
3.32	0.04	AB	–	AB	–	AA	–	AA	–	
3.29	0.03	AB	–	AB	–	AA	–	AA	–	
3.28	0.01	AB	–	AB	–	AA	–	AA	–	
3.26	0.02	AB	–	AB	–	AA	–	AA	–	
3.24	0.02	AB	–	AB	–	AA	–	AA	–	
3.23	0.01	AB	–	AB	–	AA	–	AA	–	
3.22	0.01	AB	–	AB	–	AA	–	AA	–	
3.21	0.01	AB	–	AB	–	AA	–	AA	–	
3.15	0.06	AB	–	AB	–	AA	–	AA	–	
3.15	0.00	AB	–	AB	–	AA	–	AA	–	
3.15	0.00	AB	–	AB	–	AA	–	AA	–	
3.12	0.03	AB	–	AB	–	AA	–	AA	–	
3.04	0.08	AB	–	BB	0.08	0.44	AA	–	AB	0.08	0.30	
2.88	0.16	BB	0.16	0.88	BB	–	AA	–	AB	–	
2.88	0.00	BB	–	BB	–	AA	–	AB	–	
2.86	0.02	BB	–	BB	–	AA	–	AB	–	
2.81	0.05	BB	–	BB	–	AA	–	AB	–	
2.80	0.01	BB	–	BB	–	AA	–	AB	–	
2.74	0.06	BB	–	BB	–	AA	–	AB	–	
2.60	0.14	BB	–	BB	–	AA	–	BB	0.14	0.37	
2.50	0.10	BB	–	BC	0.10	0.27	AA	–	BB	–	
2.36	0.14	BB	–	BC	–	AA	–	BB	–	
2.34	0.02	BB	–	BC	–	AA	–	BB	–	
2.29	0.05	BB	–	BC	–	AA	–	BB	–	
2.24	0.05	BB	–	BC	–	AA	–	BB	–	
2.23	0.01	BB	–	BC	–	AA	–	BB	–	
2.15	0.08	BB	–	CC	0.08	0.15	AA	–	BC	0.08	0.15	
2.13	0.02	BB	–	CC	–	AA	–	BC	–	
2.11	0.02	BB	–	CC	–	AA	–	BC	–	
2.07	0.04	BB	–		CC	–		AB	0.04	0.03	BC	–		
2.04	0.03	BB	–	CC	–	AB	–	BC	–	
2.04	0.00	BB	–	CC	–	AB	–	BC	–	
2.04	0.00	BB	–	CC	–	AB	–	BC	–	
2.00	0.04	BB	–	CC	–	BB	0.04	0.00	BC	–	
2.00	0.00	BB	–	CC	–	BB	–	BC	–	
1.78	0.22	BC	0.22	0.07	CD	0.22	0.45	BC	0.22	0.07	CC	0.22	0.23	
1.71	0.07	BC	–	CD	–	BC	–	CC	–	
1.55	0.16	CC	0.16	0.00	CD	–	CC	0.16	0.00	CC	–	
1.33	0.22	CD	0.22	0.06	DD	0.22	0.40	CD	0.22	0.06	CD	0.22	0.06	
1.27	0.06	CD	–	DD	–	CD	–	CD	–	
0.93	0.34	DD	0.34	0.00	DD	–	DD	0.34	0.00	DD	0.34	0.00	
0.37	0.56	Fail	0.56	0.19	Fail	0.56	0.19	Fail	0.56	0.19	Fail	0.56	0.19	
0.31	0.06	Fail	–	Fail	–	Fail	–	Fail	–	
0.18	0.13	Fail	–	Fail	–	Fail	–	Fail	–	
0.00 (lower bound)	0.18	–	0.18	–	–	0.18	–	–	0.18	–	–	0.18	–	

Based on Eq. (2), the outputs obtained via WGF-CPD method manifests higher Ω′, causing M-CPD method to return WGF-CPD’s result. Z-score method assigns all PVLs whereas γu is greater than 0.37, resulting in zero Ω′.

WGF-CPD:Ω′=1×1.66−0.001.66−0.00×11+0.32=0.76K-CPD:Ω′=(1−01)×1.26−0.001.66−0.00×11+0.12=0.68PAM-CPD:Ω′=1×1.58−0.001.80−0.00×11+0.46=0.60Z-score:Ω′=0×1.64−0.001.80−0.00×11+0.13=0.00.

The last data set STD2 undergoes two rounds of conditional discretization using WGF-CPD method, which eventually excludes PRL F as depicted in Table 7. Similarly, K-CPD and PAM-CPD methods re-assigns only four PRLs as illustrated in Table 7. Their Ω′ values are calculated below resulting in M-CPD method returning the CPD result from PAM-CPD. Z-score method assigns all PVLs regardless of γl greater than 23, resulting in zero Ω1.

WGF-CPD:Ω′=(1−01)×33−033−0×11+8.70=0.10K-CPD:Ω′=(1−01)×24−033−0×11+7.80=0.08PAM-CPD:Ω′=(1−01)×30−033−0×11+7.27=0.11Z-score:Ω′=0×33−040−0×11+3.11=0.00

Table 7 CPD results of STD2 data set.

Performance value (sorted)	Gap from either preceding performance value or upper bound	WGF-CPD	K-CPD	PAM-CPD	Z score	
Assigned PRL	γu, δi, γl	νi(I)	Assigned PRL	γu, δi, γl	νi(I)	Assigned PRL	γu, δi, γl	νi(I)	Assigned PRL	γu, δi, γl	νi(I)	
100 (upper bound)	–	–	–	–	–	–	–	–	–	–	–	–	–	
90	10	A	10	0	A	10	0	A	10	20	A	10	12	
90	0	A	–	A	–	A	–	A	–	
90	0	A	–	A	–	A	–	A	–	
90	0	A	–	A	–	A	–	A	–	
80	10	B	10	10	B	10	10	A	–	A	–	
80	0	B	–	B	–	A	–	A	–	
79	1	B	–	B	–	A	–	A	–	
78	1	B	–	B	–	A	–	A	–	
75	3	B	–	B	–	A	–	B	3	5	
70	5	B	–	B	–	A	–	B	–	
70	0	B	–	B	–	A	–	B	–	
70	0	B	–	B	–	A	–	B	–	
70	0	B	–	B	–	A	–	B	–	
70	0	B	–		B	–		A	–		B	–		
70	0	B	–	B	–	A	–	B	–	
57	13	C	13	20	C	13	17	B	13	7	C	13	7	
56	1	C	–	C	–	B	–	C	–	
55	1	C	–	C	–	B	–	C	–	
54	1	C	–	C	–	B	–	C	–	
53	1	C	–	C	–	B	–	C	–	
52	1	C	–	C	–	B	–	C	–	
51	1	C	–	C	–	B	–	C	–	
50	1	C	–	C	–	B		C	–	
43	7	C	–	C	–	C	7	6	D	7	6	
41	2	C	–	C	–	C	–	D	–	
40	1	C	–	C	–	C	–	D	–	
39	1	C	–	D	1	16	C	–	D	–	
38	1	C	–	D	–	C	–	D	–	
38	0	C	–	D	–	C	–	D	–	
37	1	C	–	D	–	C	–	D	–	
27	10	D	10	4	D	–	D	10	4	F	10	4	
27	0	D	–	D	–	D	–	F	–	
27	0	D	–	D	–	D	–	F	–	
26	0	D	–	D	–	D	–	F	–	
26	0	D	–	D	–	D	–	F	–	
26	0	D	–	D	–	D	–	F	–	
25	1	D	–	D	–	D	–	F	–	
24	1	D	–	D	–	D	–	F	–	
24	0	D	–	D	–	D	–	F	–	
23	1	D	–	D	–	D	–	F	–	
0
(lower bound)	23	–	23	–	–	23	–	–	23	–	–	23	–	

Outlier sensitivity analysis

Since empirical assessment generally provides performance evaluation based on limited datasets, the following analyses of algorithmic sensitivity to outliers (and noises) ensure broader generalizability across dataset variations.

Outliers are extreme performance values that significantly deviate from main data distribution. A well-designed CPD method should minimize their impact on PRL assignments while preserving fairness in ranking. WGF-CPD: Outliers create exaggerated gaps between performance values. Since WGF-CPD selects the widest gaps for segmentation, a single outlier can disrupt PRL distribution by unintentionally excluding either the first or the last assignable PRL. As outliers significantly increase the largest gaps, the method will over-segment performance values, leading to each boundary PRL covering a small or even zero PVI in a region where the outliers are present. This uneven distribution of PVIs undermines the fairness and consistency of the discretization process. Additionally, the unintended exclusion of certain PRLs reduces the granularity of the CPD, limiting its ability to accurately reflect the relative performance of individuals within the dataset.

K-CPD: This K-means-based method inherently mitigates outlier influence by forming groups around majority performance data points. Outliers can pull centroids away from the main clusters, leading to distorted PRL assignments.

PAM-CPD: Unlike K-means, which calculates centroids as means, PAM selects actual data points (medoids) as cluster centers, making it more robust to outliers and preserving the integrity of PRL assignments. This is evident in STD2 where the four performance values of 90, considered outliers, are not assigned a distinct PVL as seen in K-CPD’s results.

M-CPD: This method adaptively selects WGF-CPD, K-CPD, or PAM-CPD result with the highest Ω′ metric, inherently mitigating the impact of outliers. If outliers severely distort WGF-CPD results, M-CPD will favor either K-CPD or PAM-CPD output. However, if outliers also affect K-CPD, M-CPD will select PAM-CPD, ensuring robustness against extreme deviations.

Noise sensitivity analysis

Noisy data introduces small and random variations that affect performance values, potentially disrupting PRL assignments. WGF-CPD: Noise artificially inflates or deflates small gaps, leading to erratic PRL segmentation. WGF-CPD has low noise robustness as it directly relies on precise gap values.

K-CPD and PAM-CPD: Noise minimally affects clustering, as PRLs are assigned based on groups rather than absolute values. In other words, noise only impacts clustering when variance is high enough to shift assignments significantly. K-CPD and PAM-CPD tolerate high noises as clustering absorbs small variations. However, PAM-CPD offers an advantage over K-CPD because K-CPD assumes data points are clustered around centroids, making it less effective for irregular or non-spherical clusters, which are often treated as noise, reducing performance. In contrast, PAM-CPD handles such clusters better by minimizing the sum of dissimilarities rather than variance.

M-CPD: If noise affects WGF-CPD significantly, M-CPD selects either K-CPD or PAM-CPD as better alternatives.

Finding and discussion

We summarize the findings along with discussions based on the CPD results as follows: Illustrated in Fig. 1, K-CPD tends to contribute more significantly to the output of M-CPD compared to WGF-CPD and PAM-CPD due to higher degree of Ω′ for EMP1 and EMP2. Nevertheless, STD1 and STD2 yield the higher degrees of Ω′ when using WGF-CPD and PAM-CPD, respectively, establishing all constituent methods as indispensable components of M-CPD to function optimally. Notice that Z-score method delivers zero Ω′ for all data sets due to its failure to meet Requirement 1.

Figure 1 projects Ω′ values ranging widely among the proposed algorithms, depending on the characteristics of the sizes and distribution of performance-value divides, within each data set. The reasons for the highest and lowest degrees of conditional unbiasedness in every data set are further analyzed in Fig. 2.

Across all evaluation data sets, the Ω1 values of WGF-CPD, K-CPD, PAM-CPD, and M-CPD consistently reach their upper bound of 1.00 in Fig. 2 since the number of unassigned PRLs is sufficient to cover Θ with respect to each of the data sets. This completely achieves Requirement 1 and surpasses the performance of the conventional norm-referenced CPD method, Z score. Z-score method fails to unassign PVLs across all of the data sets resulting in zero Ω1. However, if large gaps exist between performance values within a dataset, Z-score method will demonstrate its capability for conditional discretization. For example, in STD1, if the last three performance values adjacent to the last one were changed to 0.18, Z-score method would exclude DD from assignment.

The plots of all data sets indicate that WGF-CPD method tends to outperform K-CPD and PAM-CPD methods in terms of Ω2 because WGF-CPD method determines PVIs by prioritizing the widest gaps of performance values, thus aligning more closely with Requirement 2.

According to all Ω3 values presented in Fig. 2, K-CPD method tends to excel over WGF-CPD method in fulfilling Requirement 3 as K-CPD indirectly factors in PVIs via Euclidian-distance model parameter, whereas WGF-CPD is totally blind to PVIs. For the same reason, PAM-CPD method also outperformed WGF-CPD method for Ω3 in EMP2 and STD2 data sets. Without outliers, K-CPD tends to maintain more comparable PVIs than PAM-CPD because K-CPD relies on centroids rather than medoids. Z-score method has the highest Ω3 values across all data sets as it divides the range between maximum and minimum t scores into equal intervals, leading to comparable PVIs.

Figure 1 Comparison of Ω′ resulting from each pair of method and data set.

Figure 2 Break-down comparison of Ω1 (blue), Ω2 (orange), and Ω3 (gray) among methods for EMP1 (upper left), EMP2 (upper right), STD1 (lower left), and STD2 (lower right).

Conclusion

This article proposes a CPD methodology achieving norm-referenced human-performance assessment with a focus on AI-assisted unbiasedness. The proposed method, namely M-CPD, integrates multi-models generated by novel WGF-CPD, K-CPD, and PAM-CPD methods to maximize the degree of conditional unbiasedness in CPD tasks. Initially, the definition of CPD was formalized and utilized to formulate a metric for conditional unbiasedness. The application of this metric was exemplified in details. Subsequently, WGF-CPD, K-CPD, PAM-CPD, and M-CPD were proposed with tractable time complexities and evaluated based on the open data sets from human performance domains. The empirical results and sensitivity analyses indicated that all constituent methods of M-CPD exclusively contributed to enhancing conditional unbiasedness in norm-referenced CPD. Therefore, our methods are a promising methodology for unbiasedness-centric norm-referenced CPD. Our future research endeavors will focus on extending norm-referenced CPD to multi-dimensional performance values and optimizing the scalability of each algorithm by using large-scale datasets, parallel computing frameworks, and algorithm-specific optimization techniques.

Supplemental Information

Supplemental Information 1 EMP1 Dataset: Employees evaluation for promotion consideration.

Supplemental Information 2 EMP2 Dataset: Employee performance rating correlated with salary increases.

Supplemental Information 3 STD1 Dataset: Evaluation of performance among higher education students.

Supplemental Information 4 Student performance in exams.

Supplemental Information 5 Simulation code assisting in K-means CPD (K-CPD) method and PAM (PAM-CPD) method.

The codes used in the evaluation are “Kmeans.rmp” and “PAM.rmp”, which are RapidMiner tool’s process files used for K-means and PAM clusterings, respectively. These codes are used to assist in the simulation of our proposed “K-means CPD (K-CPD) method” according to Algorithm 2 and “PAM CPD (PAM-CPD) method” according to Algorithm 3 that are prescribed in the article.

Additional Information and Declarations

Competing Interests

The authors declare that they have no competing interests.

Author Contributions

Thepparit Banditwattanawong conceived and designed the experiments, analyzed the data, authored or reviewed drafts of the article, and approved the final draft.

Masawee Masdisornchote performed the experiments, performed the computation work, prepared figures and/or tables, and approved the final draft.

Data Availability

The following information was supplied regarding data availability:

The code and data are available in the Supplemental Files.

The EMP1 dataset is available at Kaggle: https://www.kaggle.com/datasets/muhammadimran112233/employees-evaluation-for-promotion.

The EMP2 dataset is available at Kaggle: https://www.kaggle.com/datasets/uniabhi/ibm-hr-analytics-employee-attrition-performance.

The STD1 dataset is available at Kaggle: https://www.kaggle.com/datasets/csafrit2/higher-education-students-performance-evaluation.

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
