# Peer review of "Unbiased machine learning-assisted approach for conditional discretization of human performances"

_PeerJ Computer Science, doi:10.7717/peerj-cs.2804_

## Round 0.1 · original submission · Major Revisions

After a thorough review of your manuscript by the assigned reviewers, I would like to inform you that, while there is potential in your work, several significant concerns have been raised regarding the experimentation and methodology.

The reviewers have pointed out that certain aspects of the experimental setup lack sufficient clarity and justification. In particular, they believe that more detailed explanations and stronger validations are necessary to support your findings. Additionally, methodological improvements have been recommended to ensure the robustness and reliability of the results.

In light of these concerns, we are requesting major revisions to the manuscript. We kindly ask that you carefully address each of the reviewers' comments in your revised submission, providing additional detail and supporting evidence where necessary.

Reviewer 1 ·

Basic reporting

All comments have been added in detail to the last section.

Experimental design

All comments have been added in detail to the last section.

Validity of the findings

All comments have been added in detail to the last section.

Additional comments

Review Report for PeerJ Computer Science
(Unbiased machine learning-assisted approach for conditional discretization of human performances)

1. Within the scope of the study, a multi-modal technique approach including artificial intelligence for norm-referenced performance discretization is proposed.

2. In the introduction section, the importance of the subject and conditional performance discretization are mentioned in detail. However, the contributions of the study to the literature should be added to the end of this section by explaining them more clearly in items.

3. Although the Related Works section explains the literature on the subject addressed in the study at a basic level, it is definitely recommended to add a literature table to this section for deep literature analysis and to explain the differences of the study from the literature more clearly.

4. When the K-means, multi-modal and widest-gap-first conditional performance discretization methods and algorithms are examined in detail, it is observed that they are sufficiently explained and have a certain level of originality.

5. The fact that the dataset required for the analysis of the method proposed in the study is more than one and its types and content are sufficient, increases the quality of the study and the reliability of the method results.

6. When the obtained results are examined, it is understood that they are generally at an acceptable level.

In conclusion, although the study is very valuable in terms of contributing to the subject and literature, attention should be paid to the sections listed above.

·

Basic reporting

Provide more context on existing norm-referenced performance discretization methods.
Elaborate on the unsupervised machine learning method and heuristic approach.
Discuss the theoretical basis for the proposed decision function and conditional unbiasedness.
Assess the robustness of the proposed approach to outliers and noisy data.
Investigate scalability in large-scale performance evaluation applications.
Explore application in other domains.

Experimental design

Develop more advanced machine learning-based methods for performance discretization.
Conduct comprehensive performance comparison with existing methods.

Validity of the findings

Investigate potential applications in related fields (e.g., talent management, organizational development).

Additional comments

Provide more information on dataset characteristics and experimental setup.

---

## Round 0.2 · accepted · Accept

I hope this message finds you well. After carefully reviewing the revisions you have made in response to the reviewers' comments, I am pleased to inform you that your manuscript has been accepted for publication in PeerJ Computer Science.

Your efforts to address the reviewers’ suggestions have significantly improved the quality and clarity of the manuscript. The changes you implemented have successfully resolved the concerns raised, and the content now meets the high standards of the journal.

Thank you for your commitment to enhancing the paper. I look forward to seeing the final published version.

Reviewer 1 ·

Basic reporting

All comments have been added in detail to the last section.

Experimental design

All comments have been added in detail to the last section.

Validity of the findings

All comments have been added in detail to the last section.

Additional comments

Review Report for PeerJ Computer Science
(Unbiased machine learning-assisted approach for conditional discretization of human performances)

Thanks for the revision. Replies to comments and changes to the paper are sufficient. Best regards.

·

Basic reporting

The revised article have incorporated all suggested review comments.

Experimental design

All review comments are updated in the revised article.

Validity of the findings

All are addressed.

Additional comments

Nil